# PerSRV: Personalized Sticker Retrieval with Vision-Language Model

## Abstract

Instant Messaging is a popular mean for daily communication, allowing users to send text and stickers. As the saying goes, "a picture is worth a thousand words", so developing an effective sticker retrieval technique is crucial for enhancing user experience. However, existing sticker retrieval methods rely on labeled data to interpret stickers, and general-purpose Vision-Language Models (VLMs) often struggle to capture the unique semantics of stickers. Additionally, relevant-based sticker retrieval methods lack personalization, creating a gap between diverse user expectations and retrieval results. To address these, we propose the **P**ersonalized **S**ticker **R**etrieval with **V**ision-Language Model framework, namely PerSRV, structured into offline calculations and online processing modules. The online retrieval part follows the paradigm of relevant recall and personalized ranking, supported by the offline pre-calculation parts, which are sticker semantic understanding, utility evaluation and personalization modules. Firstly, for sticker-level semantic understanding, we supervised fine-tuned LLaVA-1.5-7B to generate human-like sticker semantics, complemented by textual content extracted from figures and historical interaction queries. Secondly, we investigate three crowd-sourcing metrics for sticker utility evaluation. Thirdly, we cluster style centroids based on users' historical interactions to achieve personal preference modeling. Finally, we evaluate our proposed PerSRV method on a public sticker retrieval dataset from WeChat[1], containing 543,098 candidates and 12,568 interactions. Experimental results show that PerSRV significantly outperforms existing methods in multi-modal sticker retrieval. Additionally, our fine-tuned VLM delivers notable improvements in sticker semantic understandings. The code is annoymously available[2].

## CCS Concepts

• **Information systems** → **Information retrieval**; **Personalization**.

## Keywords

Multi-modal Search, Personalization, Sticker Retrieval

---

[1]https://algo.weixin.qq.com

[2]https://anonymous.4open.science/r/persrv-5E1F

**ACM Reference Format:**

Anonymous Author(s). 2024. PerSRV: Personalized Sticker Retrieval with Vision-Language Model. In *Proceedings of The Web Conference 2025 (the WebConf 25)*. ACM, New York, NY, USA, 9 pages. https://doi.org/XXXXXXX.XXXXXXX

## 1 Introduction

As instant messaging (IM) becomes an increasingly dominant form of communication, stickers have emerged as a powerful visual tool for conveying emotions and sentiments that text alone cannot fully express. Offering a more nuanced and immediate form of interaction, stickers enrich the human communication experience. With their widespread use on platforms like WhatsApp, WeChat, and Telegram, the need for advanced sticker retrieval systems has become critical to support users in finding the right stickers efficiently.

Despite their growing significance, retrieving appropriate stickers poses several challenges. Traditional sticker retrieval systems largely depend on labels or corresponding utterances to understand sticker semantics, which creates a significant bottleneck. Furthermore, general Vision Language Models (VLMs) struggle to capture the unique, vibrant, human-like semantics of stickers. Besides, the relevant-based retrieval methods lack user preference modeling, leading to a mismatch between users' expected styles and the retrieved stickers. These limitations highlight the need for personalized sticker retrieval methods that can capture the sticker semantics and user preferences.

Other promising aspect of advancing sticker retrieval system is sticker utility metrics. Understanding a sticker's utility can enhance a sticker's retrieval accuracy. However, since sticker retrieval scenario starkly contrast with image retrieval - where the formal emphasize on emotional expression and the latter focusing on visual information, the traditional image quality evaluation metric cannot be directly applied. Popularity is a common metric used in retrieval tasks; however we aim to explore other sticker-scenario specific evaluation metrics that can effectively quantify a sticker's quality beyond mere popularity. pecifically, we brewe introduce, evaluate and compare three utility metrics—Cross User Adaptability, Sticker Popularity, and Query Adaptability.

In this paper, we present PerSRV (Personalized Sticker Retrieval with Vision Language Model), a framework integrated with multi-modal semantic understanding, sticker utility evaluation and user preference modeling. Specifically, our contributions can be summarized as follows:

- We address the Personalized Sticker Retrieval task, which has not been well studied before.
- We propose PerSRV, the first Vision-Language Model-based Personalized Sticker Retrieval method, structured into online recall and ranking processes, supported by offline modules for sticker semantic understanding, utility evaluation, and user preference modeling.

- Extensive experiments on a large-scale real-world dataset from WeChat demonstrate the significant improvements of our method, outperforming both sticker retrieval baselines and VLM-based methods. Ablation studies confirm the effectiveness of our framework designs.

## 2 Related Work

We outline related work on sticker retrieval, conversational recommendation personalized image search.

### 2.1 Sticker Retrieval

Most previous research emphasizes the importance of data for sticker retrieval. For instance, SRS [10] and PESRS [11] require corresponding utterances, while [20] relies on manually labeled emotions, sentiments, and reply keywords. CKES [4] has each sticker annotated with a corresponding emotion. During sticker creation, Hike Messenger [19] tags conversational phrases to stickers. The reliance on data presents a significant limitation, as stickers without associated information are excluded from consideration.

Gao et al. [10] uses a convolutional sticker encoder and self-attention dialog encoder for sticker-utterance representations, followed by a deep interaction network and fusion network to capture dependencies and output the final matching score. The method selects the ground truth sticker from a pool of sticker candidates, and its successor PESRS [11] enhances this by integrating user preferences. Zhang et al. [40] performs this on recommendation tasks. CKES [4] introduces a casual graph to explicitly identify and mitigate spurious correlations during training. PBR [37] paradigm enhances emotion comprehension through knowledge distillation, contrastive learning, and improved hard negative sampling to generate diverse and discriminative sticker representations for better response matching in dialogues. PEGS [39], StickerInt [23] generates sticker information using multimodal models and selects sticker responses, but does not consider personalization. Moreover, most methods are limited to selecting only one sticker at a time, a major drawback for real-world sticker scenario suggestions. To add on, these ranking methods quickly become impractical on larger datasets.

### 2.2 Personalized Image Search

Much work has been done on personalized image search. FedPAM [9] achieves personalized text-to-image retrieval through a lightweight personalized federated learning solution. Specifically, the top-k most similar text-image pairs are fetched from the private database and an attention-based module generates personalized representations. These updated representation includes client-specific information for text-to-image matching. CA-GCN [14] leverages user behavior data in a Graph Convolution Neural Network model to learn user and image embeddings simultaneously. The sparse user-image interaction data is augmented to consider similarities among images which improves retrieval performance. The main difference between Unlike personalized image search, which focuses on retrieving specific images or objects, the main purpose of stickers are to convey an emotion or provide an expression. This fundamental difference highlights the need for tailored approaches

in sticker retrieval that address emotional context rather than object recognition.

### 2.3 Image and Sticker Utility Evaluation

Sticker utility is a largely unexplored area. Unlike traditional image quality metrics such as SSIM and PSNR [24], which often focus on visual fidelity or perceived quality, a sticker's utility is measured through its ability to express emotions and expressions. Moreover, since most stickers are derived from existing images, sticker reevaluation through these methods might be insignificant.

Ge, Jing [12] performed analysis to reveal that sticker's main utility, amongst others, is to express emotions and convey behavior, action and attitude, however this is challenging to measure in stickers. Jiang et al. [15], A. T. Kariko et al. [16] indicates utilization augments personal happiness. Gygli, M et al. [13] observes interesting gifs gaining curiosity and attention, however this does not directly align with a sticker's purpose of emotional expressiveness. Constantin, Mihai Gabriel, et al. [5] deciphers the visual concepts of affective value and emotions with dimension emotion space of valence, arousal and dominance. However, these are challenging to measure. Zhang et al. [39] employ Empathy-multimodal, Consistency, and Rank as key metrics to evaluate multimodal conversational responses. However, these metrics are inherently dependent on the conversational context. In contrast, we intend to explore sticker-independent metrics.

While popularity has been a widely used metric in retrieval tasks [31], our research seeks to quantitatively assess a sticker's emotional expressiveness within the sticker-scenario.

### 2.4 Large Multimodal Models

LLMs, such as ChatGPT [1] [3], LLaMA [34], demonstrate powerful language capabilities, recent researches have extended LLMs to multimodal domains. Flamingo [2] exhibits promising zero-shot capabilities by adding a cross-attention layer. BLIP [22] [21], MiniGPT-4 [42] [41], MiniGPT-5 [41] and LLaVA [26] [25] uses a small intermediate model to bridge between the frozen vision encoder and the LLM. GILL [17] explores mapping LLM outputs into input space of vision decoder, empowering LLM's image generation capabilities. InstructBLIP [6] and UniMC [35] leverages pretrained models to enhance generalization. These recent innovations enable more effective and nuanced communication through images, generating descriptive captions that align closer to the label. However, most existing research predominantly focuses on general images.

## 3 Personalized Sticker Retrieval with VLM

In this section, we present the proposed **Per**sonalized **S**ticker **R**etrieval with **V**ision-Language Model method, abbreviated as PerSRV. We begin by introducing the problem settings in Section 3.2, followed by an overview of the framework in Section 3.1. The framework's offline pre-calculation modules are then detailed in Section 3.3, 3.4, 3.5, and the online components are introduced in Section 3.6.

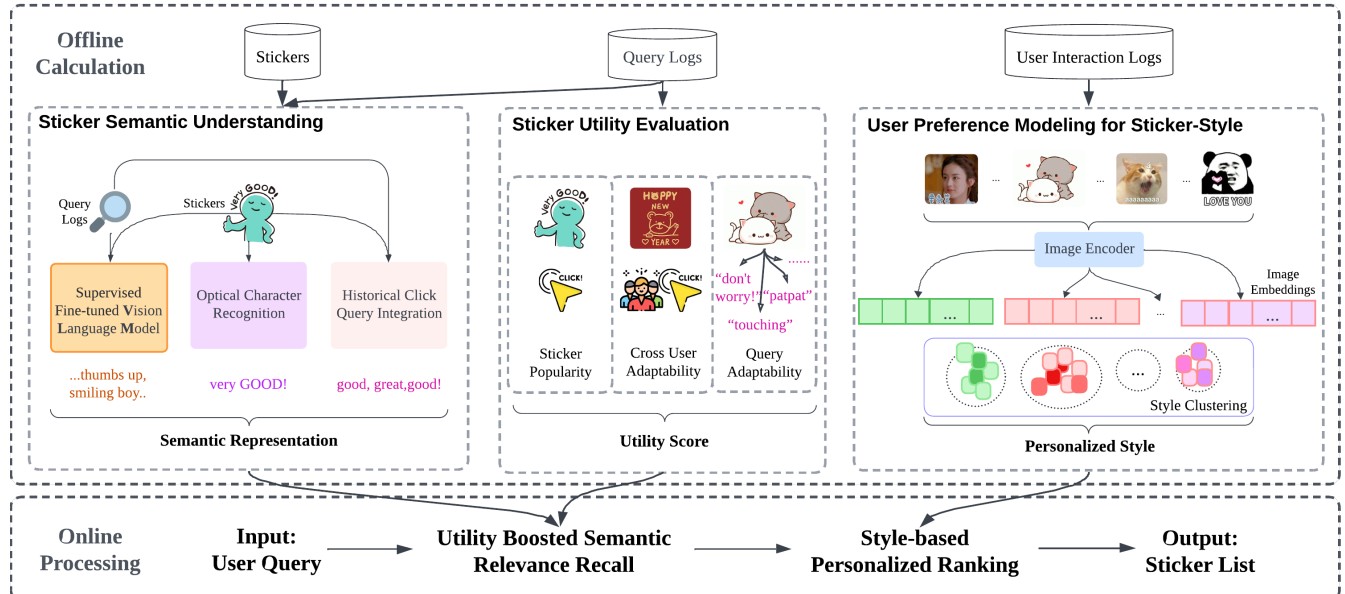

**Figure 1: Overview of PerSRV. The framework is structured into offline calculation and online processing. The offline preparation has three modules: (1) *Multi-modal Sticker Semantic Understanding*; (2) *Sticker Utility Evaluation*; (3) *User Preference Modeling for Sticker-Style*. When an online query comes, PerSRV first recall semantically relevant, utility enhanced stickers and then rank according to user preference.**

## 3.1 Framework Overview

## 3.2 Problem Formulation

We address the standard personalized retrieval task in sticker scenarios. Given users' historical query logs $\{(l_{q1}, l_{u1}, l_{s1}), .., (l_{qL}, l_{uL}, l_{sL})\}$, a query $q$ from user $u$ and a set of candidate stickers $S = \{s_1, \ldots, si\}$, the objective is to provide a selected rank list of stickers, prioritizing positive examples near the top.

As shown in Figure 1, the *Personalized Sticker Retrieval with Vision-Language Model* framework is structured into offline pre-calculation and online processing stages. The offline stage consists of three key modules: Sticker Semantic Understanding, Sticker Utility Evaluation, and User Preference Modeling, discussed in Section 3.3, 3.4, 3.5. These pre-calculated features further feed into the online phase. In the online stage, PerSRV follows a standard two-step process of recall and ranking. First, it recalls utility-boosted semantic relevance stickers, then applies personalized style-based ranking. The details of this process are explained in Section 3.6.

## 3.3 Multi-modal Semantic Understanding

A critical aspect of semantic understanding is generating accurate, human-like descriptions for stickers, an area where previous sticker retrieval approaches and general VLMs fall short of. To achieve a more comprehensive semantic understanding, we leverage (1) Supservised Fine-tuned Visional Language Model to generate human-like keywords, (2) Optical Character Recognition (OCR) to capture textual content in stickers and (3) Historical Click Qury Information for context integration. These three techniques are detailed below.

*3.3.1 Vision-Language Model for Sticker Understanding.* To modeling the overall semantics of stickers, we utilize Vision Language Model (VLM), specifically Llava-1.5-7b [26] for sticker understandings. Initially, we test the Llava's in-context capabilities to generate human-like keywords using the following instruction:

---

**Instruction for VLM to obtain sticker semantics**

*You are a sticker expert. Please carefully observe and understand the meanings that the sticker wants to convey and give a brief phrase or two that expresses the semantic meanings of this sticker.*

---

However, experiments reveal that Llava alone is insufficient for capturing the nuanced, human-like keywords associated with stickers. These results are discussed further in Section 5.3. To improve performance, we supervised fine-tuned the VLM using Low-Rank Adaptation (LoRA) on the training set, with human click queries serving as ground truth for generating sticker keywords.

Additionally, We agument these keywords with general image captions, including image description and the emotions evoked by the sticker, using the following format for the general VLM.

*This is a sticker. 1 Pease describe the sticker in detail. 2 describe the feelings it is meant to express. Use this format 1. | 2.*

*3.3.2 Optical Character Recognition for Textual Content Extraction.* A significant attribute of many stickers is the inclusion of text within the image, which provides essential cues for accurately capturing semantic meaning.

We employ Optical Character Recognition (OCR), a well-established technique for extracting text from images [32]. Specifically, we use the open-source PaddleOCR model [7] to extract textual information from the stickers. For each candidate sticker, the model identifies any present text along with its recognition confidence score. After manual inspection, text with a confidence score above 0.7 is retained. After the ORC process, 73% (398,066 out of 543,098) of the stickers obtain the textual information in the image.

*3.3.3 Query Integration from Historical Interactions.* In retrieval scenarios, historical query logs offer an important dimension for capturing item semantics from real users in a collaborative filtering manner. To fully utilize the rich interaction logs and enhance the human-like semantic understanding of stickers, we integrate the associated queries corresponding to each sticker in the training set into the semantic understanding of stickers. This integration reflects real users' contextual understanding, thereby enriching the semantic representation of the stickers.

## 3.4 Sticker Utility Evaluation

Beside semantic relevance requirements, another key factor for real-world retrieval systems is to provide high-quality results.

We investigate three crowd-sourcing driven factors for utility modeling. They are sticker popularity, cross user adaptability and query adaptability.

*3.4.1 Sticker Popularity.*

$$Pop_s = |[c \in \text{Logs} : \text{clicks } c \text{ for sticker } s]| \quad (1)$$

*3.4.2 Cross User Adaptability.*

$$CrossUserAdapt_s = |\{u \in \text{Users} : \text{user } u \text{ clicked on sticker } s\}| \quad (2)$$

*3.4.3 Query Adaptability.*

$$QueryAdapt_s = |\{q \in Q : \text{query } q \text{ matches } s\}| \quad (3)$$

The final score is calculated as follow,

$$\mathbf{U} = \begin{bmatrix} Pop_s + \text{base} \cdot \mathbb{1}[Pop_s = 0] \\ CrossUserAdapt_s + \text{base} \cdot \mathbb{1}[CrossUserAdapt_s = 0] \\ QueryAdapt_s + \text{base} \cdot \mathbb{1}[QueryAdapt_s = 0] \end{bmatrix} \quad (4)$$

Then, the utility score can be expressed as,

$$Utility_s = \mathbf{w}^T \cdot \sqrt{Norm(\mathbf{U})} \quad (5)$$

## 3.5 User Preference Modeling for Sticker Styles

Personalization is a key factor in enhancing user experience within online systems [8]. In the PerSRV framework, personalization is achieved by modeling users' preferred sticker styles, such as sticker series and included elements.

We utilize the pretrained image encoder, CLIP-cn [38], to extract 512x512 embedding representations of sticker images. Then, the k-means clustering method [18] is applied to the embedding sets of each user's interacted stickers to identify style preferences. Each cluster contains a set of embeddings with a corresponding centroid. This centroid serves as the basis for the online style-based presonlaized ranking introduced in Section 3.6.2.

Case studies in Section 5.4.2 illustrate that this method could effective model user preference on sticker styles from interaction

history and experiments in Section 5.2 verifies that PerSRV's personalization significantly improve the downstream retrieval performances.

## 3.6 Online Sticker Retrieval enhanced with Offline Support

**Table 1: PerSRV Notations**

| Symbol | Description |
|---|---|
| $Semantics_s$ | Semantics of sticker s |
| $Utility_s$ | Utility Evaluation of sticker s |
| Recall Score$_q(s)$ | Recall Score for sticker s under query q |
| $Style_u$ | Preferred Style Cluster for user u |
| Preference Score$_u(s)$ | Preference Score of user u for sticker s |
| Score$_{u;q}(s)$ | Ranking Score of user u, sticker s and query q |

To support timely retrieval from the large number of candidates, we use a two-step approach when online queries comes. They are the utility boosted semantic relevance recall and the style-based personalized ranking.

*3.6.1 Utility Boosted Semantic Relevance Recall.* PerSRV's first aim is to recall relevant stickers for search queries. At the same time, we further add the sticker utility boosted factor in the recall phrase to ensure user experience. We use BM25 [30] to calculate the semantic relevance score between query and stickers. This is supported with the offline prepared sticker semantics from SFT VLM keywords, OCR texts and integrated queries. Then, the pre-calculated sticker-level utility score, defined in the earlier section, is integrated to enhance the recall process. Based on the above recall score, we get the top R (100) stickers from the large candidate sets.

$$\text{Recall Score}_q(s) = Relevance(q; Semantics_s) + Utility_s \quad (6)$$

$$\text{where } Semantics_s = \{VLM_s, OCR_s, QueryInteg._s\}, \quad (7)$$

$$Utility_s = \{Pop_s, CrossUserAdapt_s, QueryAdapt_s\}. \quad (8)$$

In the above equations, $q$ is the online query and $s$ represent each candidate stickers.

*3.6.2 Stype-based Personalized Ranking.* After recalling the semantic relevance and high-utility stickers, we further prepare the ranked list with personalization, where we use the offline calculated user preferred style clustering.

User preference for each sticker is defined as the shortest distance between the sticker's embedding and the centroids of the user's style preferences, as calculated in Section 3.5. The preference score is computed according to the following equation:

$$\text{Preference Score}_u(s) = \min_{Style_u} Distance(Style_u, s) \quad (9)$$

Finally, we combine both the recall score and the preference score to compute the final score, which is used to rank the final stickers:

$$\text{Score}_{u;q}(s) = \text{Recall Score}_q(s)(1 + \alpha \text{Preference Score}_u(s)) \quad (10)$$

This approach allows for personalized ranking by factoring in both semantic relevance, high-utility and user style preferences.

# 4 Experimental Setup

## 4.1 Dataset

We introduce the public dataset from WeChat [36] with sticker as response in detail.

**Table 2: Statistics of the WeChat dataset. The dataset has 543,098 stickers and 12,568 user-query-sticker logs.**

| Field | Number |
|---|---|
| # Stickers | 543,098 |
| # User-Query-Sticker | 12,568 |
| # Unique User-Query Pairs | 2,308 |
| # Unique Queries | 1,891 |

We utilize the public large-scale user query interaction dataset with stickers from one of the most popular messaging apps. As shown in Table 2, there are 543,098 stickers, 12,569 interactions, and 8 users. We randomly extract 80% of the interactions for training and reserve the remaining 20% for testing and validation.

## 4.2 Evaluation Metrics

Following the challenge [36], we employ multi-mean reciprocal ranking M-MMR@$k$ as an evaluation metric, which measures the relative ranking position of positive responses. Following a previous study [10], we also employ recall $R@k$ as an evaluation metric, which measures if the positive responses are ranked in the top $k$ candidates.

## 4.3 Baseline Details

- **Global Pop, User Pop**: [31] count the occurrence of most common stickers globally and user-level to generate a list of stickers.
- **BM25**: [30] ranks documents based on the relevance of term frequency and inverse document frequency.
- **BM25 (+OCR)**: [30] performs the same as before but includes OCR text.
- **BLIP2**: [21] aligns visual and textual information through efficient pre-training. We translated the English captions into Chinese using Opus-MT-en-zh [33].
- **CLIP-cn**: [38] uses contrastive learning to align images and text in a shared embedding space. We input sticker images and collect 512x512 image embeddings.
- **SRS**: [10] matches sticker image features with multi-turn dialog context using convolutional and self-attention encoders. The open-source model EmojiLM [29] generate the required emoji annotations using our captured semantics.

We conducted our experiments using PyTorch [28] on an NVIDIA A100-SXM4 GPU.

# 5 Experimental Result

In the following subsections, we aim to answer the following questions:

- **RQ1**: What is the overall performance of PerSRV compared with all baselines?
- **RQ2**: What is the effect of each module in PerSRV?
- **RQ3**: How effective is VLM in sticker comprehension?
- **RQ4**: How effective are proposed sticker quality metrics?

## 5.1 Overall Results

For research question **RQ1** , we evaluated the performance of our model alongside various baseline methods across multiple metrics, as detailed in Table 3. We split our baselines into popularity, text, general-VLM, and sticker-based methods to conduct a comprehensive comparison of our proposed method.

The popularity-based approaches [31] demonstrated limited effectiveness, with the Global Popularity method achieving a maximum M-MRR@20 score of only 0.0015, while the User Popularity method fared slightly better at 0.0028 for M-MRR@20. Similarly, general VLM-based approaches [21] [38] yielded modest results, with CLIP-cn reaching just 0.0072 at M-MRR@5. In contrast, the SRS method [10] showed moderate performance, attaining an M-MRR@20 score of 0.0087. Notably, text-based approaches exhibited significantly higher performance, with the BM25 (Query+OCR) method [30] achieving an impressive M-MRR@20 of 0.2772.

Our proposed method, PerSRV, outperformed all baseline methods, achieving an outstanding M-MRR@20 of 0.3020—approximately **8.95%** higher than the second-best method—and demonstrating a remarkable **19%** improvement in M-MRR@1. This validates the effectiveness of method on the sticker retrieval task.

## 5.2 Ablation Study

In this section, we address **RQ2**. Specifically, we remove personalization component from the framework, as shown in Table 3. Removing the Personalization Ranking resulted in substantial performance drops across several key metrics, with M-MRR@10 and Recall@10 decreasing by approximately 14.58%, M-MRR@20 decreasing by 9.06%, Recall@20 decreasing by 6.30%, and M-MRR@1 showing a decrease of 7.32%. This underscores the critical role of personalization in sticker retrieval, demonstrating that tailored approaches significantly enhance user engagement and satisfaction, leading to enhanced retrieval outcomes.

## 5.3 Analysis on Vision-Language Model

Quantitatively, we assess the performance of the visual language model (VLM) through two key evaluations. We compare the keywords generated by both the base VLM and the SFT VLM.

We utilize the BLEU [27] score to evaluate the proximity of the generated keywords to the ground truth. To investigate OCR text influence on the SFT model, we differentiated between stickers with and without OCR text; this is indicated by the two extra columns: *BLEU w/o OCR* and *BLEU w/ OCR*. Column *w/o OCR text* indicates BLEU score evaluation on sticker candidates without OCR text and column *w/ OCR Text* indicates BLEU score evaluation on sticker candidates with OCR text. As shown in Table 4, our SFT model exhibits a remarkably higher BLEU score across all categories. Specifically, the SFT model demonstrates an astounding performance increase of approximately 1.2 billion folds compared to the base model, indicating a significant improvement in keyword generation quality.

To complement the BLEU score, which may not fully capture the semantic relationships between words, we employ cosine similarity

**Table 3: Baseline Evaluation Comparison. Our proposed method, PerSRRV, achieved the highest performance among all evaluated methods. Text-based approaches ranked second, while general VLM and popularity-based methods exhibited the lowest performance. The statistical significance of differences observed between the performance of two runs is tested using a two-tailed paired t-test and is denoted using * for significance at $\alpha = 0.05$ and ** for significance at $\alpha = 0.01$. We bold the best results and underline the second best.**

| | | M-MRR@1 | R@1 | M-MRR@5 | R@5 | M-MRR@10 | R@10 | M-MRR@20 | R@20 |
|---|---|---|---|---|---|---|---|---|---|
| Popularity-based | Global Pop | 0.0000 | 0.0000 | 0.0009 | 0.0006 | 0.0013 | 0.0012 | 0.0015 | 0.0024 |
| | User Pop | 0.0011 | 0.0011 | 0.0022 | 0.0015 | 0.0025 | 0.0027 | 0.0028 | 0.0037 |
| Text-based | BM25 | 0.0097 | 0.0097 | 0.0418 | 0.0364 | 0.0494 | 0.0535 | 0.0494 | 0.0535 |
| | BM25 (+OCR) | 0.0852 | 0.0852 | 0.2414 | 0.1804 | 0.2682 | 0.2269 | 0.2772 | 0.2726 |
| General VLM | BLIP2 | 0.0000 | 0.0000 | 0.0005 | 0.0003 | 0.0011 | 0.0014 | 0.0012 | 0.0021 |
| | CLIP-cn | 0.0021 | 0.0021 | 0.0072 | 0.0060 | 0.0078 | 0.0087 | 0.0087 | 0.0123 |
| Sticker Ranking | SRS | 0.0064 | 0.0015 | 0.0086 | 0.0051 | 0.0102 | 0.0096 | 0.0115 | 0.0175 |
| Ours | PerSRV | **0.1014**** | **0.1014**** | **0.2673**** | **0.1889**** | **0.2938**** | **0.2326**** | **0.3020**** | **0.2736**** |
| | *w/o Personal.* | 0.0885 | 0.0885 | 0.2451 | 0.1777 | 0.2722 | 0.2265 | 0.2814 | 0.2740 |

**Table 4: BLEU score comparison between SFT and base VLM. The fine-tuned model demonstrates an astounding performance increase across all categories. As the BLEU score from the base model is relatively insignificant, (i.e. $3 \times 10^{-10}$), we have omitted it from the table.**

| Model | BLEU Overall | BLEU w/o OCR Text | BLEU w/ OCR Text |
|---|---|---|---|
| Base | 0.0000 | 0.0000 | 0.000 |
| SFT | **0.3625**** | **0.3829**** | **0.3580**** |

**Table 5: Cosine similarity comparison between SFT and base VLM. The SFT VLM performs significantly better in both keyword generation scenarios with and without OCR texts.**

| Model | CosSim Overall | CosSim w/o OCR Text | CS w/ OCR Text |
|---|---|---|---|
| Base | 0.3756 | 0.3757 | 0.3755 |
| SFT | **0.6863**** | **0.6979**** | **0.6837**** |

as an additional evaluation metric. Similarly, to investigate the OCR text influence on the SFT model, we differentiated between stickers with and without OCR text; this is indicated by the two extra columns: *CosSim w/o OCR Text* and *CosSim w/ OCR Text*. Column *CosSim w/o OCR Text* indicates cosine similarity score evaluation on sticker candidates without OCR text and column *CosSim w/ OCR Text* indicates cosine similarity score evaluation on sticker candidates with OCR text.

Table 5 illustrates the cosine similarity scores for both the base and SFT VLM. The SFT VLM shows a substantial increase in similarity both with and without utilizing OCR text; specifically SFT VLM demonstrates an approximate **80%** increase across different categories.

Collectively, the results from both the BLEU and cosine similarity metrics emphasize that the SFT VLM demonstrates a strong quantitative ability to generate effective sticker keywords, particularly for the large amount of stickers lacking labels or utterances.

Lastly, we showcase our method's semantic understanding of stickers. Figure 2 demonstrates the semantic understanding capabilities of both the base and SFT VLM for a sticker with and without OCR text. The figure highlights the multi-layered progression of semantic comprehension, starting from shallow layers like general descriptions to emotions elicitation, to more precise, contextually rich keywords generated by the SFT model. The blue box are description and emotions elicited from the VLM. As shown in Figure 2, the base model is unable to generate an accurate prediction whereas the VLM model is able to precisely generate the keywords, as indicated by the green tick.

Moreover, the SFT model demonstrates its effectiveness by accurately predicting the actual keywords, despite it being different than the OCR text. This showcases its robustness independence of the OCR text. This underscores the SFT VLM's capability to improving the retrieval process by generating more accurate and semantically relevant keywords.

## 5.4 Case Study

*5.4.1 Vision-Language Model.* We present concrete examples from our dataset that highlight the improvements achieved with the PerSRV model. Figure 3 shows a sticker retrieval case focusing on image emotion and description prompting. First, we prompt the VLM to generate both sticker descriptions and associated emotions. The VLM's response is placed in the dark blue container, with the relevant responses highlighted in orange and light blue. Secondly, stickers are retrieved based on the generated prompt. Without sticker description and emotions context, the query "Want to cry, but no tears come" fails to retrieve the ground truth sticker. However, the inclusion of VLM output retrieves the ground truth sticker successfully in the first position. This underscores the importance

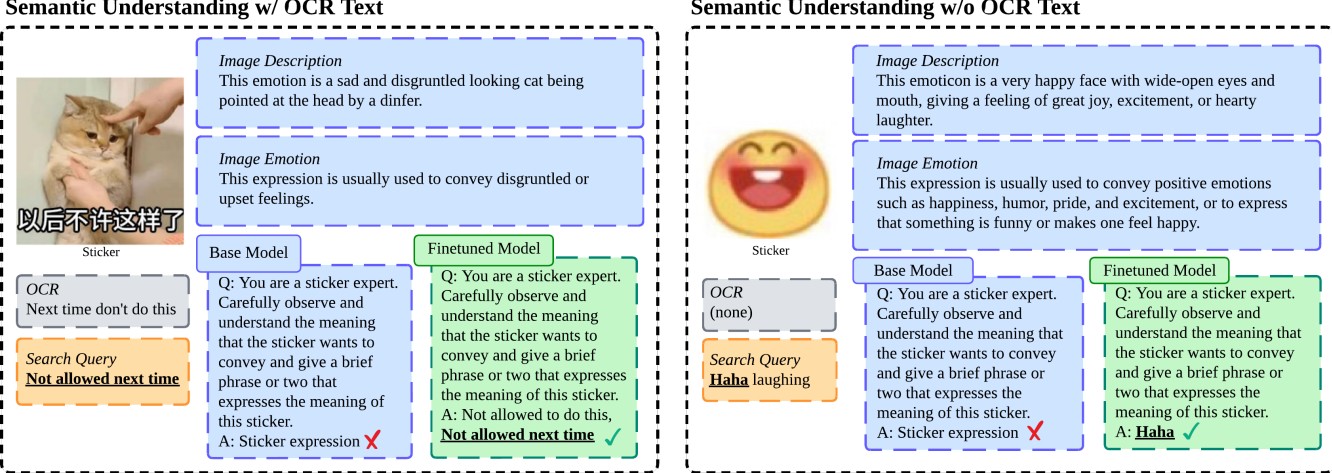

**Figure 2: Example of semantic understanding for stickers with and without OCR text. The SFT VLM is capable of accurately generating keywords. The green box indicates the SFT VLM's response and the green tick indicates an accurate prediction to the ground truth stickers.**

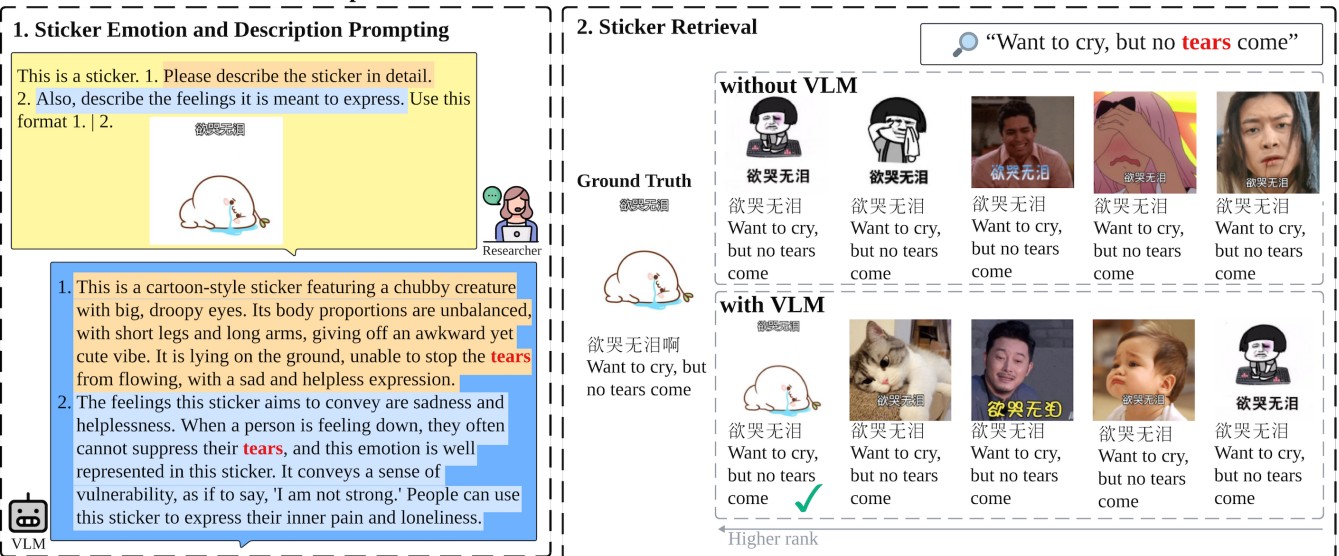

**Figure 3: Example of sticker description and emotions prompting. The inclusion of description and emotions allows PerSRV to recall and rank the ground truth significantly higher, which was previously not observed.**

of offline image description generation and emotion elicitation for more accurate and efficient sticker retrieval.

*5.4.2 Personalization.* In Fig 4, we show the significance of personalization in downstream task. By considering User 1's preferences, we observe a marked increase in the relevance of retrieved sticker candidates, which are closely aligned with the user's ground truths.

*5.4.3 Quality Score.* Fig 5 highlights the impact of sticker utility metrics on sticker retrieval. The top row is retrieval without quality score and the bottom row integrates quality score. In this instance, the ground truth improved its ranking from the 4th position to the 1st position, as indicated by the green arrow, demonstrating the effectiveness of the quality score metrics generated by PerSRV. This significant enhancement in relevance underscores the capability of the quality score to elevate the most pertinent sticker candidates.

**Sticker Personalization**

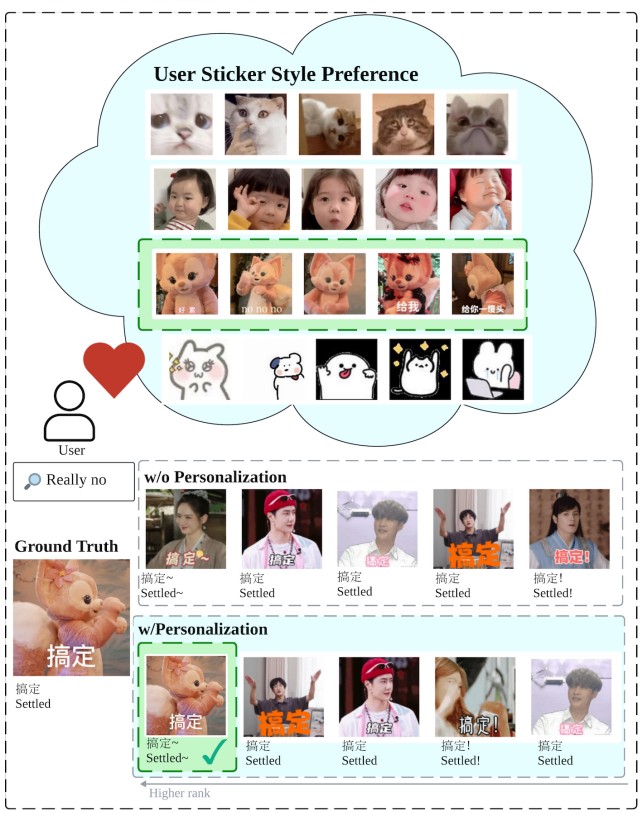

**Figure 4: Example of sticker retrieval with user style preference. User prefers a classic Disney character; in the retrieval task, this preference is considered.**

**Quality Score**

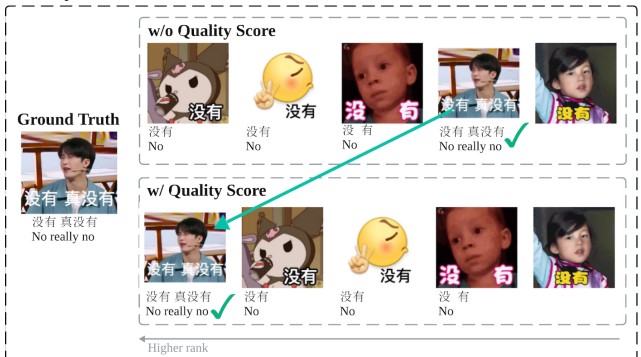

**Figure 5: Example of quality score ablation generated by PerSRV, illustrating the effectiveness of the quality score metrics.**

# 6 Discussion

## 6.1 Inference Time Evaluation

**Table 6: Evaluation of inference times for different baselines.**

|  | M-MRR@10 | Inference Time/query (s) |
|---|---|---|
| BLIP2 + Opus | 0.0010 | 0.0549 |
| CLIP-cn | 0.0068 | 0.0914 |
| SRS | 0.0102 | 0.1103 |
| PerSRV | **0.2722** | **0.0541** |

Table 6 presents the evaluation of inference times across various baselines, highlighting the efficient performance of PerSRV compared to traditional methods. Notably, while other models like SRS exhibit significantly longer inference times, PerSRV maintains rapid query processing with minimal latency, crucial for ensuring a smooth user experience in real-time sticker retrieval applications.

## 6.2 Extensibility

The PerSRV framework demonstrates notable extensibility, demonstrating a robust solution for sticker retrieval applications. In contrast to conventional systems, which rely heavily on labeled data or user utterances—two significant bottlenecks in retrieval tasks—our method utilizes Vision-Language Models (VLMs). This approach effectively obviates the need for such requirements, thereby streamlining the deployment process and mitigating potential data concerns by minimizing the dependence on sensitive user information. Moreover, PerSRV's method emphasizes efficient data management, as the majority of data preparation occurs offline. The method also requires minimal training time, which enhances its adaptability across various user contexts and applications.

# 7 Conclusion

In this work, We address the Personalized Sticker Retrieval task, which has not been well studied before. We propose PerSRV, the first Vision-Language Model-based Personalized Sticker Retrieval method, structured into online recall and ranking processes, supported by offline modules for sticker semantic understanding, utility evaluation, and user preference modeling. Extensive experiments on a large-scale real-world dataset from WeChat demonstrate the significant improvements of our method, outperforming both sticker retrieval baselines and VLM-based methods. Ablation studies confirm the effectiveness of our framework designs.

Furthermore, we emphasize the role of personalization in enhancing user experience, tailoring sticker retrieval to individual preferences and thereby improving overall satisfaction. Through comprehensive experiments, we demonstrate the practicality and effectiveness of our system, achieving significantly better performance compared to existing methods.

We believe that our findings can pave the way for further sticker retrieval work. Our work not only contributes to the existing body of knowledge but also lays the groundwork for future advancements in personalized sticker retrieval works.

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
