# OpenReview forum: "PerSRV: Personalized Sticker Retrieval with Vision-Language Model"
_ACM.org/TheWebConf/2025/Conference — WWW 2025 Oral_

### Official Review · Reviewer_2CG2 · 2024-12-01

**Novelty:** 5
**Technical Quality:** 5

**Review:**

The paper addresses a significant practical issue in modern communication platforms, enhancing sticker retrieval relevance and personalization, thereby substantially improving user experience. Furthermore, the framework’s extensibility holds promising potential for applications in other domains.

Strengths:
1. Innovative Framework: combines vision-language models, utility evaluation, and user preference modeling to propose a novel solution for personalized sticker retrieval.
2. Robust Personalization Support: effectively incorporates user preferences, significantly enhancing retrieval performance.

Weakness:
1. Limitations of Static Assumptions: assuming user preferences are static may not align with the dynamic nature of real-world user behaviors.
2 . Insufficient Explanation of Technical Details: some design choices, such as parameter settings or model selection, are not adequately justified.
3. Can the use of K-means clustering effectively capture complex user behaviors or dynamic preferences?
4. Have the proposed utility metrics (e.g., popularity, cross-user adaptability, query adaptability) been validated through user experiments?

**Questions:**

see above

**Reviewer Confidence:**

3: The reviewer is confident but not certain that the evaluation is correct

**Scope:**

3: The work is somewhat relevant to the Web and to the track, and is of narrow interest to a sub-community

---

### Official Review · Reviewer_jYWt · 2024-12-02

**Novelty:** 4
**Technical Quality:** 6

**Review:**

This paper proposes a stick retrieval model (i.e., PerSRV), which consists of three modules, Sticker Semantic Understanding, Sticker Utility Evaluation, and User Preference Modeling. The authors first extract SFT VLM, OCR, and historical query features to construct semantic representations. They then incorporate sticker popularity, cross-user adaptability, and query adaptability to enhance performance. Style-based personalization is also employed to refine the scoring. The results show that the proposed PerSRV outperforms previous methods by a large margin.

Strengths:
1. The paper is well-organized and straightforward. The tables and case studies are rich and easy to understand.

2. Ablation studies and analysis demonstrate how personalized features affect the final performance and how OCR text influences the VLM's performance.

3. The final performance is remarkable compared to baseline methods, providing valuable knowledge about sticker retrieval.

Weaknesses:
1. The proposed method appears to be more of an engineering solution than an insightful research work. All the techniques used have been proven effective in many other studies.

2. The column names in Tables 4 and 5 are misleading. Do the terms 'w/ OCR' and 'w/o OCR' refer to stickers that contain text or not? It's easy to misunderstand them as an ablation study evaluating the effectiveness of OCR.

3. Important ablation studies are lacking. How would PerSRV be affected if features from the VLM, OCR, Query Integration, or the terms in the utility evaluation were removed?

4. The URL to the repo is active but contains nothing.

**Questions:**

This paper originates from the authors’ proposed solution in the Wechat sticker retrieval competition. However, could the authors consider conducting additional experiments on other publicly available datasets, such as StickerInt [1] or MCDSCS [2]? This will enable a comprehensive assessment of PerSRV's capabilities, particularly of its generalization across different datasets.

[1] Liang, Bin, et al. "Reply with Sticker: New Dataset and Model for Sticker Retrieval." arXiv preprint arXiv:2403.05427 (2024).

[2] Shi, Yuanchen, and Fang Kong. "Integrating Stickers into Multimodal Dialogue Summarization: A Novel Dataset and Approach for Enhancing Social Media Interaction." Proceedings of the 32nd ACM International Conference on Multimedia. 2024.

**Reviewer Confidence:**

3: The reviewer is confident but not certain that the evaluation is correct

**Scope:**

4: The work is relevant to the Web and to the track, and is of broad interest to the community

---

### Official Review · Reviewer_6R1H · 2024-12-02

**Novelty:** 4
**Technical Quality:** 3

**Review:**

This paper presents PerSRV, a personalized sticker retrieval framework that enhances sticker search by combining multimodal semantic understanding, utility-based retrieval, and personalized style modeling. It uses Vision-Language Models (VLMs), Optical Character Recognition (OCR), and historical query logs to improve sticker semantics, with fine-tuned VLMs generating more accurate human-like keywords. The framework introduces three utility metrics—sticker popularity, cross-user adaptability, and query adaptability—to assess sticker quality and enhance retrieval accuracy. Personalization is achieved by clustering user interaction data to model style preferences, which are integrated into the retrieval process. PerSRV employs a two-step approach in online retrieval: first, utility-boosted semantic relevance recall, followed by personalized style-based ranking, ensuring both semantic relevance and user-specific preferences are addressed.
Pros:
1.The paper presents a promising approach to personalized sticker retrieval with its PerSRV framework.
2.The experimental results demonstrate the effectiveness of the model.
3.The analysis of the case studies is thorough, and the selected cases effectively illustrate the sticker retrieval process.

Cons:
1.The article contains formatting and spelling errors:
(1)The section, which is supposed to provide an overview of the framework, only contains the title and lacks any explanation, leaving a significant gap in understanding the system's structure and methodology.
(2)Meanwhile, take table 3 as an example, there is an alignment issue in  the "w/o Personal." row.
(3)Additionally, there are numerous inappropriate line spacing throughout the paper. The spacing between subsections-subsections/paragraph and figures/tables-paragraph should be uniformly adjusted.
(4)Furthermore, the illustration in Figure 2 should be placed below the image, not above it.
2.The paper lacks a thorough analysis of the baselines, such as StickerInt[1], StickerCLIP and StickerLLM[2], and the ablation study does not sufficiently demonstrate the importance of each component of the model.
3.The ablation study in the paper is limited, as it only investigates the personalization component of the proposed framework. Key parts of the framework, such as the Sticker Utility Evaluation components and the Historical Click Query Integration in the Sticker Semantic Understanding module, are not included in the ablation analysis. While these aspects are discussed and partially validated in the case study, case studies typically demonstrate results on specific instances rather than providing evidence for the generalizability and robustness of the individual components across different contexts. This raises concerns about the thoroughness of the ablation study and its ability to adequately validate the full model.
4.The experimental dataset only includes WeChat. If there are other publicly available sticker retrieval datasets from different social platforms, they could help us better validate the effectiveness and generalizability of the proposed method.
5.While the paper claims that the code is publicly available, upon checking the provided link, it is found that the code is still marked as "coming soon," which undermines the transparency and reproducibility of the proposed approach.
[1]Liang B, Wang B, Bai Z, et al. Reply with Sticker: New Dataset and Model for Sticker Retrieval[J]. arXiv preprint arXiv:2403.05427, 2024.
[2]Zhao S, Ge Y, Qi Z, et al. Sticker820k: Empowering interactive retrieval with stickers[J]. arXiv preprint arXiv:2306.06870, 2023.

**Questions:**

Q1. Some modules of the proposed model lacks for adequate experimental validation and analysis. For example, how does the sticker utility evaluation module and the historical click query integration improve the performance?
Q2. Is it sufficient to evaluate the proposed method using only the WeChat dataset, or is it possible to find publicly available sticker retrieval datasets other than WeChat for experimental validation?
Q3. The dataset used in the experiments appears to be quite limited, with only 8 users mentioned in Section 4.1. Given the small user pool, will the results may lack generalizability to larger or more diverse populations, or is this a typo?
Q4. Why is the code not yet available despite being stated as publicly accessible?

**Reviewer Confidence:**

3: The reviewer is confident but not certain that the evaluation is correct

**Scope:**

3: The work is somewhat relevant to the Web and to the track, and is of narrow interest to a sub-community

---

### Official Review · Reviewer_9UeM · 2024-12-02

**Novelty:** 6
**Technical Quality:** 6

**Review:**

This paper presents a novel framework for enhancing sticker retrieval in instant messaging applications. By leveraging Vision-Language Models and focusing on personalization, the authors address significant limitations in existing sticker retrieval systems, which often rely on labeled data and lack user preference modeling.

Pros:
1. It introduces a unique approach to sticker retrieval by integrating multi-modal semantic understanding, utility evaluation, and user preference modeling. This comprehensive framework enhances the retrieval process significantly compared to traditional methods.
2. The introduction of specific utility metrics for stickers, such as Cross User Adaptability and Query Adaptability, provides a more nuanced understanding of sticker quality beyond mere popularity. This innovation addresses a significant gap in existing research.

Cons:
1.  If the training data lacks sufficient diversity, the model may inadvertently reinforce existing biases within the dataset. This could result in homogeneous recommendations that fail to accommodate the diverse tastes and preferences of a global audience, ultimately leading to a less inclusive user experience

**Questions:**

The same as above.

**Reviewer Confidence:**

3: The reviewer is confident but not certain that the evaluation is correct

**Scope:**

3: The work is somewhat relevant to the Web and to the track, and is of narrow interest to a sub-community